# The influence of nucleus accumbens shell D1 and D2 neurons on outcome-specific Pavlovian instrumental transfer

Octavia Soegyono, Elise Pepin, Beatrice K Leung, Billy Chieng, Bernard W Balleine, Vincent Laurent*

Decision Neuroscience Laboratory, School of Psychology, Sydney, Australia

## eLife Assessment

This study provides novel and **convincing** evidence that both dopamine D1 and D2 expressing neurons in the nucleus accumbens shell are crucial for the expression of cue-guided action selection, a core component of decision-making. The research is systematic and rigorous in using optogenetic inhibition of either D1- or D2-expressing medium spiny neurons in the NAc shell to reveal attenuation of sensory-specific Pavlovian-Instrumental transfer, while largely sparing value-based decision on an instrumental task. The **important** findings in this report build on prior research and resolve some conflicts in the literature regarding decision-making.

**\*For correspondence:**
v.laurent@unsw.edu.au

**Competing interest:** The authors declare that no competing interests exist.

**Abstract** The nucleus accumbens shell (NAc-S) and its projections to the ventral pallidum (VP) are thought to be critical for stimulus-based decisions. The NAc-S is predominantly composed of spiny projection neurons (SPNs) that express either the dopamine D1 (D1-SPNs) or the dopamine D2 receptor (D2-SPNs). Yet, the role of these two neuronal subpopulations and their inputs to the VP in stimulus-based decisions remains unknown. Here, we used optogenetics in female and male knock-in rats to selectively silence D1- or D2-SPNs and their projections to the VP at a time when the rats were required to use predictive stimuli to choose between two instrumental actions. Silencing either population of NAc-S SPNs disrupted choice. Silencing NAc-S D1-SPNs terminals in the VP also disrupted choice. However, choice was left intact by silencing NAc-S D2-SPNs terminals in the VP. Together, these findings provide novel insights into the cellular mechanisms and circuitry underlying stimulus-based decisions. We discuss how these insights are consistent with a recent model proposing that these decisions are controlled by an opioid-based memory system residing in the NAc-S.

## Introduction

Our choices and decisions are often influenced by predictive signals available in the environment (*Hollis, 1984*). This influence is studied in the laboratory using the outcome-specific Pavlovian-instrumental transfer (PIT) task (*Trapold and Overmier, 1972*; *Colwill and Rescorla, 1988*; *Holmes et al., 2010*; *Cartoni et al., 2016*; *Laurent and Balleine, 2021*; *Leung et al., 2024b*), which comprises three stages. The first two stages are Pavlovian and instrumental conditioning, which can be administered in any order. In Pavlovian conditioning, two stimuli (S1 and S2; e.g., sounds or lights) are paired with two distinct and motivationally significant outcomes (O1 and O2; e.g., food outcomes). In instrumental conditioning, subjects are trained to perform two actions (A1 and A2), with each earning one of the two outcomes (i.e., A1 → O1 and A2 → O2). The final stage is the PIT test, which evaluates choice between the two actions in the presence (and absence) of the stimuli. Typically, each of the

Pavlovian stimuli biases choice toward the action with which it shares an outcome: that is, S1: A1 > A2 and S2: A2 > A1.

At a neural level, PIT has been found to rely on interactions between a number of structures, including cortical (*Keistler et al., 2015*; *Bradfield et al., 2015*; *Bradfield et al., 2018*; *Tensaouti et al., 2025*; *Ostlund and Balleine, 2007*; *Lichtenberg et al., 2021*; *Sias et al., 2021*; *Lichtenberg et al., 2017*), amygdala (*Corbit and Balleine, 2005*; *Morse et al., 2020*; *Sias et al., 2024*; *Lichtenberg et al., 2017*; *Shiflett and Balleine, 2010*; *Derman et al., 2020*; *Prévost et al., 2012*), basal ganglia (*Corbit and Balleine, 2011*; *Corbit et al., 2001*; *Corbit and Janak, 2010*; *Morse et al., 2020*; *Laurent et al., 2012*; *Bertran-Gonzalez et al., 2013*; *Laurent et al., 2014*; *Corbit et al., 2016*; *Leung et al., 2024a*; *Leung and Balleine, 2015*; *Leung and Balleine, 2013*; *Laurent et al., 2015*; *Prévost et al., 2012*), midbrain (*Corbit et al., 2007*; *Sias et al., 2024*; *Seitz et al., 2022*), and thalamic (*Ostlund and Balleine, 2008*; *Leung et al., 2024a*) territories. Among these, the nucleus accumbens shell (NAc-S) stands out because, unlike the others, it does not itself contribute to learning the stimulus–outcome (S–O) or the action–outcome (A–O) associations formed during the Pavlovian and instrumental stages of the PIT task and appears only to mediate their interaction (*Corbit and Balleine, 2011*; *Corbit et al., 2001*; *Morse et al., 2020*; *Laurent et al., 2012*; *Laurent et al., 2014*). Therefore, it has been proposed that the NAc-S acts as a central hub integrating S–O and A–O information at the time of PIT to guide choice between actions in an outcome-selective manner (*Shiflett and Balleine, 2010*; *Bertran-Gonzalez and Laurent, 2018*; *Laurent and Balleine, 2021*). Accordingly, PIT is disrupted by manipulations undermining NAc-S function during the PIT test (*Morse et al., 2020*; *Corbit et al., 2016*; *Laurent et al., 2012*; *Laurent et al., 2014*; *Laurent et al., 2015*).

The NAc-S is predominantly composed of spiny projection neurons (SPNs), which can be classified into two distinct subpopulations depending on the dopamine receptor they express (*Gerfen and Surmeier, 2011*). One population harbors the dopamine D1 receptors (D1Rs; D1-SPNs), whereas the other population expresses the dopamine D2 receptors (D2Rs; D2-SPNs). To date, only one study has examined whether activity in both populations is recruited during PIT (*Laurent et al., 2014*), finding that PIT is associated with an increase in ERK1/2 phosphorylation in D1-SPNs but not D2-SPNs. Further, blockade of D1Rs in NAc-S during choice between actions disrupted PIT whereas D2Rs blockade had no effect. Although these findings indicate a major involvement of NAc-S D1-SPNs in PIT, caution must be exerted when drawing conclusions on the role of D2-SPNs. While D1Rs are exclusively expressed on NAc-S D1-SPNs, D2Rs are present on cholinergic interneurons (CINs) and local presynaptic dopamine terminals in addition to D2-SPNs (*Gerfen and Surmeier, 2011*). Additionally, research shows that ERK1/2 phosphorylation may not capture all transcriptional events that occur in D2-SPNs (*Bertran-Gonzalez et al., 2008*; *Matamales et al., 2020*). Lastly, pharmacological D2Rs blockade has been found to be quite ineffective at influencing D2-SPNs function (*Tozzi et al., 2007*). Thus, the relative contribution of NAc-S D1- and D2-SPNs in driving choice between actions during PIT remains unclear.

The present experiments, therefore, aimed to provide an unambiguous assessment of the roles played by the two main populations of NAc-S SPNs during choice between actions in an outcome-specific PIT task. This assessment was achieved through optogenetic silencing in two knock-in rat lines that express Cre recombinase in either D1- or D2-SPNs (*Pettibone et al., 2019*). We first used tract-tracing and ex vivo electrophysiology to confirm our capacity to selectively silence activity in NAc-S D1- or D2-SPNs. We then conducted two experiments that assessed outcome-specific PIT while D1- or D2-SPNs in the NAc-S were silenced during presentations of the predictive stimuli in the choice test. Finally, we assessed the degree to which the influence of the two SPNs populations on outcome-specific PIT depends on their projections to the ventral pallidum (VP). We focused on the VP for two main reasons: First, both NAc-S D1- and D2-SPNs send dense projections to this region (*Lu et al., 1997*; *Kupchik et al., 2015*); and, second, communication between the NAc-S and VP has previously been found to be critical for outcome-specific PIT (*Leung and Balleine, 2013*).

# Results

## Anterograde tracing and ex vivo cell recordings in D1-Cre and A2a-Cre rats

We first examined the connectivity of striatal D1- and D2-SPNs in transgenic rats expressing Cre recombinase in neurons harboring the dopamine D1 receptor (D1-Cre rats) or the adenosine A2a receptor (A2a-Cre rats) (*Pettibone et al., 2019*). A2a receptors are predominantly expressed on striatal D2-SPNs (*Schiffmann et al., 2007*), thereby enabling selective targeting of these SPNs in the A2a-Cre line. D1-Cre (*n* = 1 female and 1 male) and A2a-Cre rats (*n* = 1 female and 1 male) were unilaterally infused with a Cre-dependent eYFP virus in the NAc-S (*Figure 1A*). To validate expression in these rat lines, other D1-Cre (*n* = 4 males) and A2a-Cre rats (*n* = 3 males) were given a unilateral infusion in the dorsal striatum (DS) with the same virus. In both transgenic strains, viral expression was largely restricted to local SPNs in the two targeted regions: ~95% of eYFP-positive neurons expressed DARPP-32 (D32 | eYFP; *Figure 1C–F*). Viral expression was consistent with D1- and D2-SPNs representing two neuronal populations similar in size: in both transgenic strains and targeted regions ~40% of DARPP-32-positive neurons expressed eYFP (eYFP | D32; *Figure 1C–F*). DS infusion in D1-Cre rats resulted in eYFP-positive terminals (*Figure 1B, C*) in the substantia nigra pars reticulata (SNr) and the globus pallidus externus (GPe). By contrast, the same infusion in A2a-Cre rats only produced eYFP-positive terminals in the GPe (*Figure 1B, D*). NAc-S infusion in both D1-Cre and A2a-Cre rats resulted in eYFP-positive terminals in the VP (*Figure 1E, F*). By contrast, eYFP-positive terminals in the lateral hypothalamus (*Figure 1E, F*) were only observed in D1-Cre rats. Overall, these results are in line with current knowledge about the cellular composition of the striatum (*Gerfen and Surmeier, 2011*; *Bertran-Gonzalez et al., 2010*) and well-established projection patterns of striatal D1- and D2-SPNs (*Gerfen and Surmeier, 2011*; *Lu et al., 1997*; *Kupchik et al., 2015*; *O'Connor et al., 2015*). We were therefore confident in the capacity of the two rat transgenic lines to reveal the function of NAc-S D1- and D2-SPNs.

Next, we used ex vivo electrophysiological recordings to assess our capacity to silence activity of NAc-S D1- and D2-SPNs. D1-Cre and A2a-Cre rats were bilaterally infused in the NAc-S with either a null Cre-dependent eYFP virus (D1-Cre: 2 females; A2a-Cre: 2 females) or a Cre-dependent inhibitory halorhodosipin (eNpHR3.0; D1-Cre: 2 females; A2a-Cre: 2 females) virus (*Figure 2A, B*). LED light illumination (625 nm, continuous) of either NAc-S D1-SPNs (*Figure 2A*) or NAc-S D2-SPNs (*Figure 2B*) transfected with the eYFP virus had no effect on action potential firing (Baseline vs. LED ON; D1-Cre: p = 1, 5 cells; A2A-Cre: p = 0.35, 8 cells). By contrast, the same illumination onto NAc-S D1- (*Figure 2A*) or D2-SPNs (*Figure 2B*) transfected with eNpHR3.0 suppressed action potential firing (D1-Cre: $F_{(1,6)}$ = 16.82; $\eta^2$ = 0.74, p < 0.001, 7 cells; A2A-Cre: $F_{(1,5)}$ = 135.00; $\eta^2$ = 0.96, p < 0.001, 6 cells). Importantly, there was no difference in firing prior to or following LED light illumination, indicating that the optical manipulation did not alter D1- or D2-SPNs physiology. Together, these results validate the efficacy of eNpHR3.0 to specifically silence D1- and D2-SPNs activity at the time of LED light delivery.

## NAc-S D1-SPNs mediate outcome-specific PIT

This experiment examined the effects of silencing NAc-S D1-SPNs on outcome-specific PIT. D1-Cre rats were bilaterally infused in the NAc-S with either the null Cre-dependent eYFP virus (eYFP: 5 females and 5 males) or the Cre-dependent halorhodopsin (eNpHR3.0: 5 females and 5 males) virus and were implanted with fiber-optic cannulas above the NAc-S (*Figure 3A*, *Figure 3—figure supplement 1A, B*). Using the outcome-specific PIT protocol depicted in *Figure 3B*, the rats first received Pavlovian conditioning during which two stimuli (S1 and S2; white noise or clicker) were paired with two food outcomes (O1 and O2; grain pellets or sucrose solution). Next, the rats underwent instrumental conditioning and learned that pressing one lever earned one of the food outcomes whereas pressing a second lever delivered the other outcome. Finally, an outcome-specific PIT test was administered and assessed choice between the two lever press actions in the presence or absence of each stimulus. Critically, half of the trials for each stimulus took place under LED light stimulation (625 nm, continuous; ON trials) whereas the other half of the trials was conducted without the stimulation (OFF trials). Thus, NAc-S D1-SPNs were silenced on half of the trials in rats infused with eNpHR3.0.

Pavlovian and instrumental conditioning proceeded as expected (*Figure 3—figure supplement 1C, D*). The data of most interest are those from the outcome-specific PIT test and are presented in *Figure 3C* (see also *Figure 3—figure supplement 1E, F* and Appendix 1). They are plotted as the

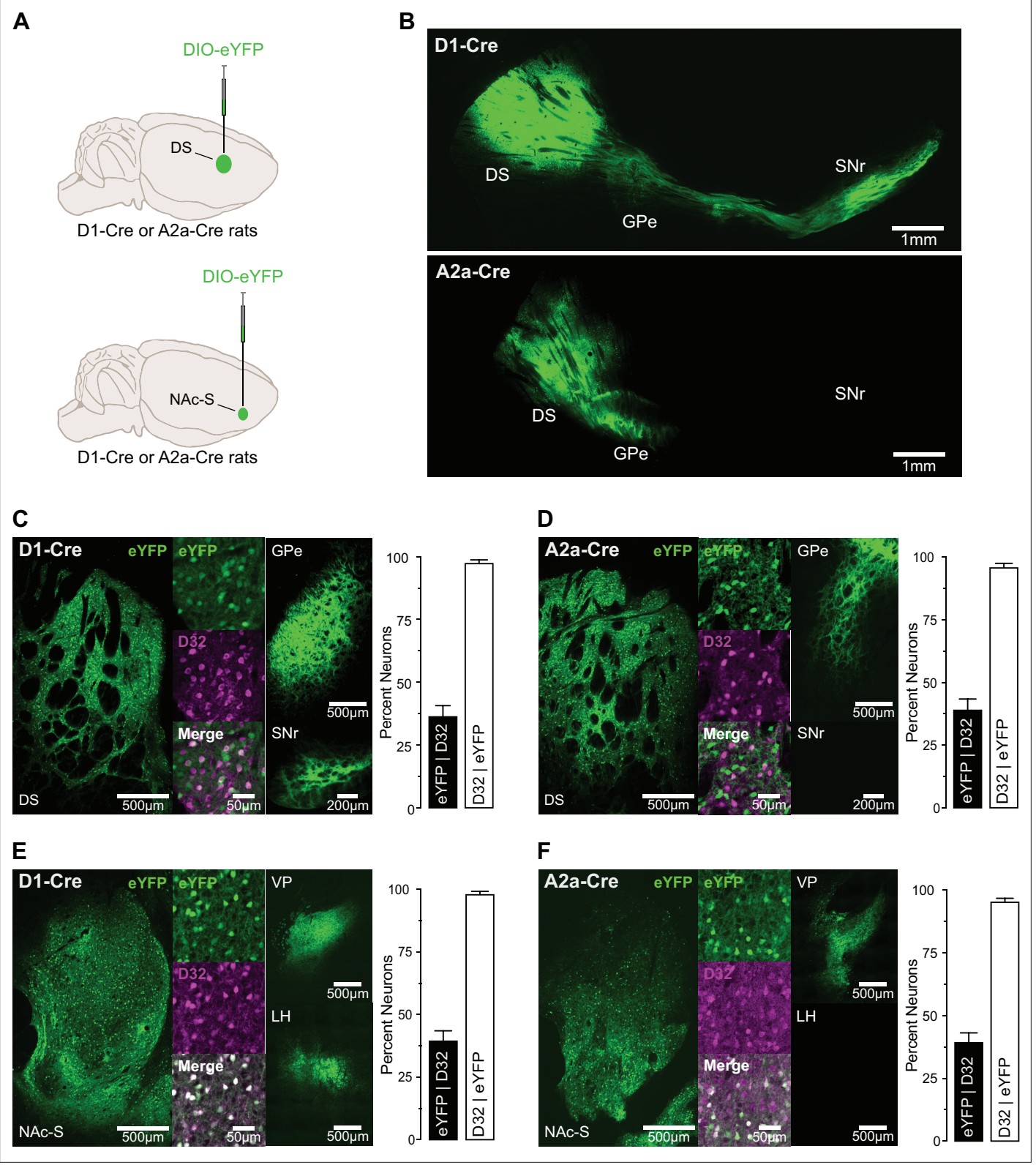

**Figure 1.** Anterograde tracing in D1-Cre and A2a-Cre rats. (**A**) D1-Cre or A2a-Cre rats were unilaterally infused in the dorsal striatum (DS; D1-Cre: *n* = 4 males; A2A-Cre: *n* = 3 males) or nucleus accumbens shell (NAc-S; D1-Cre: *n* = 1 female and 1 male; A2a-Cre: *n* = 1 female and 1 male) with DIO-eYFP. (**B**) Sagittal micrographs obtained in D1-Cre (top) and A2A-Cre (bottom) rats following viral infusion in the DS. (**C**) DIO-eYFP infusion in the DS of D1-Cre rats. Micrographs show eYFP expression in the DS, D32 staining, and co-labeling (eYFP + D32) in the DS. They also show that DS D1-SPNs project to the

*Figure 1 continued on next page*

*Figure 1 continued*

substantia nigra pars reticulata (SNr) and the globus pallidus externus (GPe). Viral expression was restricted to putative SPNs (D32 | eYFP), with ~40% of SPNs expressing eYFP (eYFP | D32). (**D**) DIO-eYFP infusion in the DS of A2a-Cre rats. Micrographs show eYFP expression in the DS, D32 staining, and co-labeling (eYFP + D32) in the DS. They also show that DS D2-SPNs project to the SNr but not the GPe. Viral expression was restricted to putative SPNs (D32 | eYFP), with ~40% of SPNs expressing eYFP (eYFP | D32). (**E**) DIO-eYFP infusion in the NAc-S of D1-Cre rats. Micrographs show eYFP expression in the NAc-S, D32 staining, and co-labeling (eYFP + D32) in the DS. They also show that NAc-S D1-SPNs project to the ventral pallidum (VP) and the lateral hypothalamus (LH). Viral expression was restricted to putative SPNs (D32 | eYFP), with ~41% of SPNs expressing eYFP (eYFP | D32). (**F**) DIO-eYFP infusion in the NAc-S of A2a-Cre rats. Micrographs show eYFP expression in the NAc-S, D32 staining, and co-labeling (eYFP + D32) in the DS. They also show that NAc-S D2-SPNs project to the VP but not the LH. Viral expression was restricted to putative SPNs (D32 | eYFP), with ~40% of SPNs expressing eYFP (eYFP | D32).

mean number of lever presses per minute when the stimulus predicted the same outcome as the action (Same) and when the stimulus predicted a different outcome to the action (Different). Thus, A1 was identified as 'Same' and A2 as 'Different' in the presence of S1. Conversely, A2 was identified as 'Same' and A1 as 'Different' in the presence of S2. Further, baseline responding (number of lever presses per minute on the two actions during the 2 min preceding each stimulus presentation) was subtracted from the rates of responding during the stimuli since it did not differ between groups (Group – eYFP vs. eNpHR3.0: p = 0.42). This approach allowed us to focus on the net effect of the stimuli on choice performance.

Silencing NAc-S D1-SPNs eliminated outcome-specific PIT. Overall lever press rates were similar between groups (Group: p = 0.48), LED light condition (LED – ON vs. OFF: p = 0.14), and the two factors did not interact (Group x Light: p = 0.07). However, lever press rates were higher on the action earning the same outcome as the stimuli compared to the action earning the different outcome (Lever – Same vs. Different: $F_{(1,18)}$ = 50.06; $\eta^2$ = 0.74; p < 0.001), regardless of group (Group x Lever: p = 0.77). There was no Lever by LED light condition interaction (Lever x LED: p = 0.72) but critically, there was an interaction between Group, LED light condition, and Lever during the presentation of the predictive stimuli (Group x LED x Lever: $F_{(1,18)}$ = 7.19; $\eta^2$ = 0.29; p = 0.02). Follow-up analyses revealed that rats in the eYFP group displayed intact outcome-specific PIT whether the LED light was OFF ($F_{(1,9)}$ = 6.14; $\eta^2$ = 0.67; p < 0.05) or ON ($F_{(1,9)}$ = 39.46; $\eta^2$ = 0.81; p < 0.001). By contrast, rats in the eNpHR3.0 group displayed outcome-specific PIT when the light was OFF ($F_{(1,9)}$ = 12.38; $\eta^2$ = 0.58; p < 0.01) but not when it was ON (p = 0.67). Together, these results indicate that activity in NAc-S D1-SPNs is necessary for the outcome-specific influence exerted by predictive stimuli on choice between actions. This is consistent with previous pharmacological work showing that NAc-S D1Rs blockade removes specific PIT (*Laurent et al., 2014*). Importantly, the impairment produced by silencing NAc-S D1-SPNs was restricted to the influence of predictive stimuli on choice between actions. In the same rats, we found that this silencing had no effect on value-based choice (*Figure 3—figure supplement 1G* and Appendix 2) using an outcome devaluation procedure: all rats performed the action that previously earned a valuable outcome more than the action that earned a devalued outcome. This agrees with studies reporting that NAc-S lesion or inactivation spares value-based choice (*Corbit and Balleine, 2011*; *Corbit et al., 2001*; *Morse et al., 2020*; *Laurent et al., 2012*; *Laurent et al., 2014*).

## NAc-S D2-SPNs mediate outcome-specific PIT

The removal of outcome-specific PIT under silencing of NAc-S D1-SPNs reproduced the impairment previously observed with blockade of NAc-S D1Rs (*Laurent et al., 2014*). Interestingly, the latter pharmacological study did not find any effect following NAc-S infusion of a D2Rs antagonist. To evaluate this finding, the present experiment examined the effects of silencing NAc-S D2-SPNs on outcome-specific PIT. A2a-Cre rats were bilaterally infused in the NAc-S with either the null Cre-dependent eYFP virus (eYFP: 4 females and 4 males) or the Cre-dependent halorhodospin (eNpHR3.0: 4 females and 2 males) virus and were implanted with fiber-optic cannulas above the NAc-S (*Figure 4A*, *Figure 4—figure supplement 1A, B*). The rats then received the outcome-specific PIT protocol previously described (*Figure 3B*).

Pavlovian and instrumental conditioning went as expected (*Figure 4—figure supplement 1C, D*). The data from the outcome-specific PIT test are presented in *Figure 4B*, again with baseline subtracted since it did not differ between groups (Group: p = 0.89; see also *Figure 4—figure supplement 1E, F*). Silencing NAc-S D2-SPNs activity eliminated outcome-specific PIT. Overall

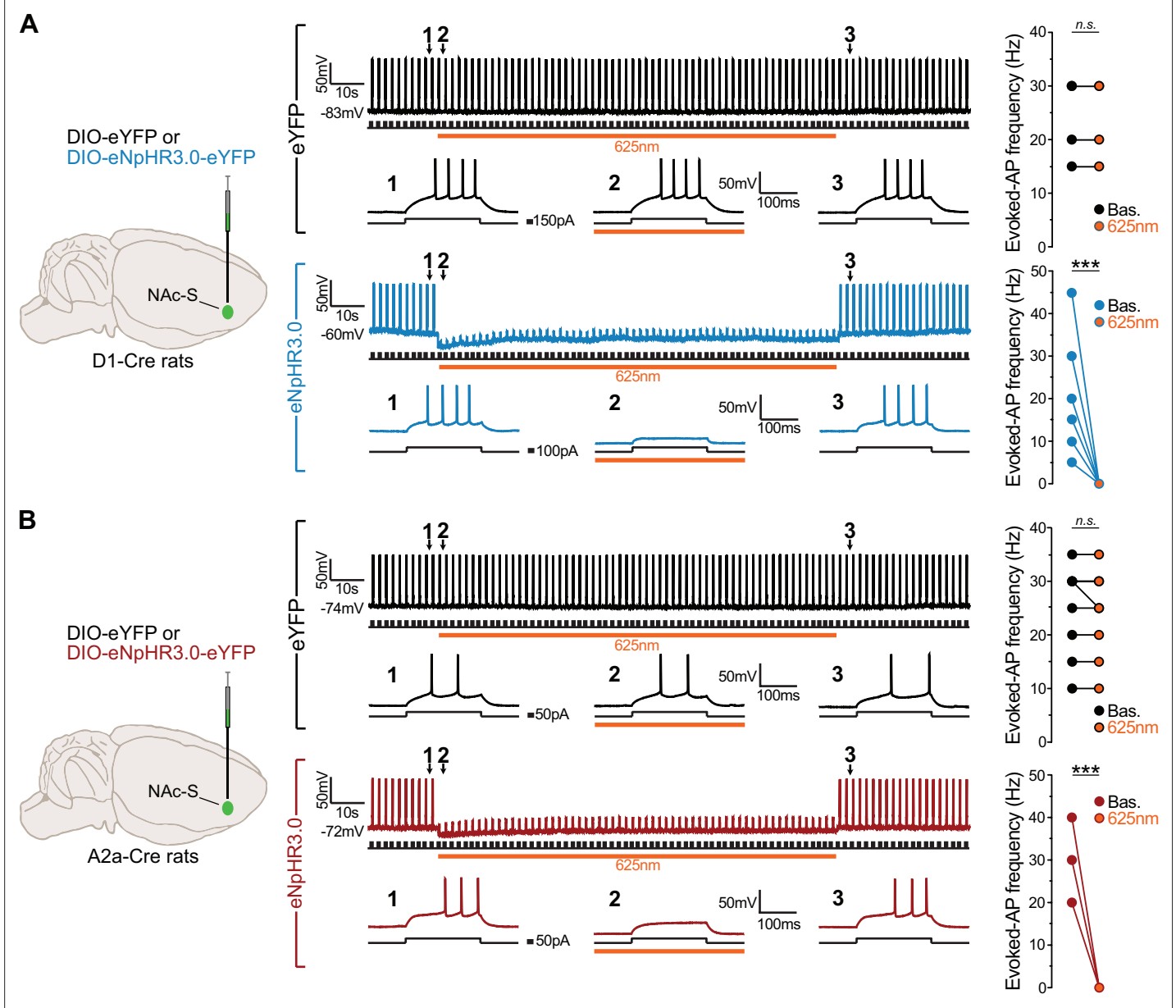

**Figure 2.** Ex vivo cell recordings in D1-Cre and A2a-Cre rats. (**A**) D1-Cre was bilaterally infused in the NAc-S with DIO-eYFP (black; 2 females) or DIO-eNpHR3.0 (blue; eNpHR3.0; 2 females). The representative raw traces of cell-attached recordings are those of transfected neurons that were depolarized to elicit action potentials by injecting a brief positive current step (+150 pA, 200 ms duration, 0.5 Hz). 625 nm LED illumination (orange bar, continuous wave, 2 mW) had no effect in eYFP transfected neurons (black; 5 cells) but it inhibited action potential in eNpHR3.0 transfected neurons (blue; 7 cells). The grouped data for recordings include overlapping data points. (**B**) A2a-Cre was bilaterally infused in the NAc-S with DIO-eYFP (black; 2 females) or DIO-eNpHR3.0 (red; 2 females). The representative raw traces of cell-attached recordings are those of transfected neurons that were depolarized to elicit action potentials by injecting a brief positive current step (+150 pA, 200ms duration, 0.5 Hz). 625 nm LED illumination (orange bar, continuous wave, 2 mW) had no effect in eYFP transfected neurons (black; 8 cells) but it inhibited action potential in eNpHR3.0 transfected neurons (red; 6 cells). The grouped data for recordings include overlapping data points.

lever press rates during the test were similar between groups (Group: p = 0.86), LED light condition (LED: p = 0.21), and the two factors did not interact (Group x LED: p = 0.70). Lever press rates were higher on the action earning the same outcome as the stimuli compared to the action earning the different outcome (Lever: $F_{(1,12)}$ = 22.77; $\eta^2$ = 0.66; p < 0.001), regardless of group (Group x Lever: p = 0.46). There was no Lever by LED light condition interaction (Lever x LED: p = 0.15) but critically, there was an interaction between group, LED light condition, and Lever during

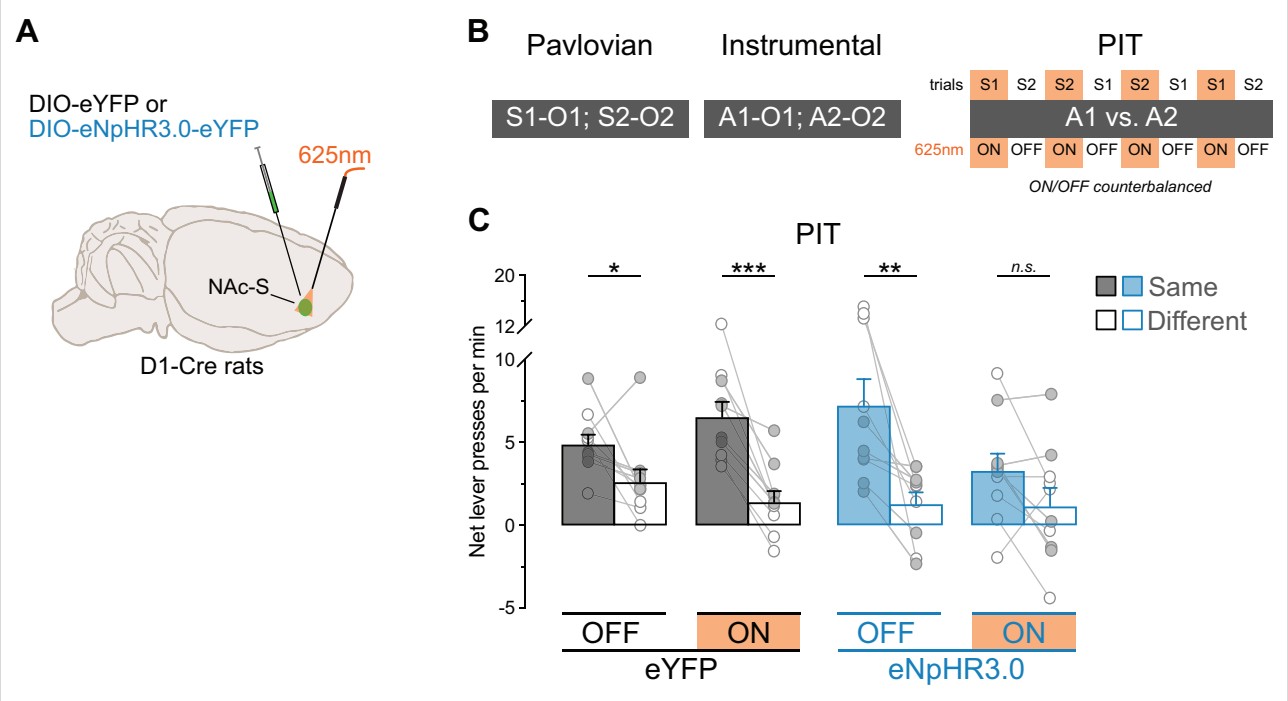

**Figure 3.** NAc-S D1-SPNs mediate outcome-specific Pavlovian instrumental transfer (PIT). (**A**) D1-Cre rats were bilaterally infused in the NAc-S with DIO-eYFP (black; 5 females and 5 males) or DIO-eNpHR3.0 (blue; 5 females and 5 males). Fiber-optic cannulas were implanted above the NAc-S to provide 625 nm LED illumination (continuous). (**B**) Schematic representation of the behavioral design; S1 and S2: noise and clicker stimuli (counterbalanced); O1 and O2: grain pellets and sucrose solution (counterbalanced); A1 and A2: left and right lever press (counterbalanced). At the test, S1 and S2 were presented four times each, in a pseudorandom order. Half of the trials for each stimulus was conducted under 625 nm LED illumination (ON; continuous wave; ~10 mW) whereas the LED remained inactivated during the other half of the trials (OFF). ON/OFF trials were counterbalanced. (**C**) Outcome-specific PIT test: net lever presses when the stimuli predicted the same outcome as the action (Same) or when the stimuli predicted the different outcome (Different). Lever presses are shown for each group in trials conducted under 625 nm LED illumination (ON) and in trials without illumination (OFF). Data are shown as mean ± SEM. Panel C includes individual data points for female (filled circle) and male (open circle) rats. Asterisks denote significant effect (*p < 0.05; **p < 0.01; ***p < 0.001; *n.s.*, nonsignificant).

The online version of this article includes the following figure supplement(s) for figure 3:

**Figure supplement 1.** Histological and behavioral data related to *Figure 3*.

the presentation of the predictive stimuli (Group LED x Lever: $F_{(1,12)}$ = 8.73; $\eta^2$ = 0.42; p < 0.01). Follow-up analyses revealed that control eYFP rats expressed outcome-specific PIT whether the LED light was OFF ($F_{(1,7)}$ = 16.25; $\eta^2$ = 0.70; p < 0.01) or ON ($F_{(1,7)}$ = 20.58; $\eta^2$ = 0.75; p < 0.01). By contrast, rats in the eNpHR3.0 group displayed outcome-specific PIT when the light was OFF ($F_{(1,5)}$ = 6.94; $\eta^2$ = 0.58; p < 0.05) but not ON (p = 0.34). Thus, these results show for the first time that activity in NAc-S D2-SPNs is necessary for the outcome-specific influence exerted by predictive stimuli on choice between actions.

The present finding contrasts with our previous observation that outcome-specific PIT remains unaffected by NAc-S infusion of a D2Rs antagonist (*Laurent et al., 2014*). However, the capacity of such pharmacological manipulation to reveal the function of D2-SPNs is unclear. For instance, a D2Rs antagonist could be expected to enhance D2-SPN activity, since D2Rs are Gi coupled and so their activation should reduce adenylyl cyclase activity. Further, D2Rs are not exclusively expressed on this cell population (*Gerfen and Surmeier, 2011*), as they can be found on local CINs and presynaptic dopamine terminals. Finally, previous work found that pharmacological treatment targeting D2Rs can leave D2-SPNs activity unaffected (*Tozzi et al., 2007*). These issues are overcome by the present optical manipulation in A2a-Cre rats, and we are therefore confident that NAc-S D2-SPNs activity is in fact critical for outcome-specific PIT. Importantly, we also found that the impairment produced by silencing NAc-S D2-SPNs was restricted to the influence of predictive stimuli on choice between actions. In the same rats, this silencing had no effect on value-based choice (*Figure 4—figure supplement 1G*).

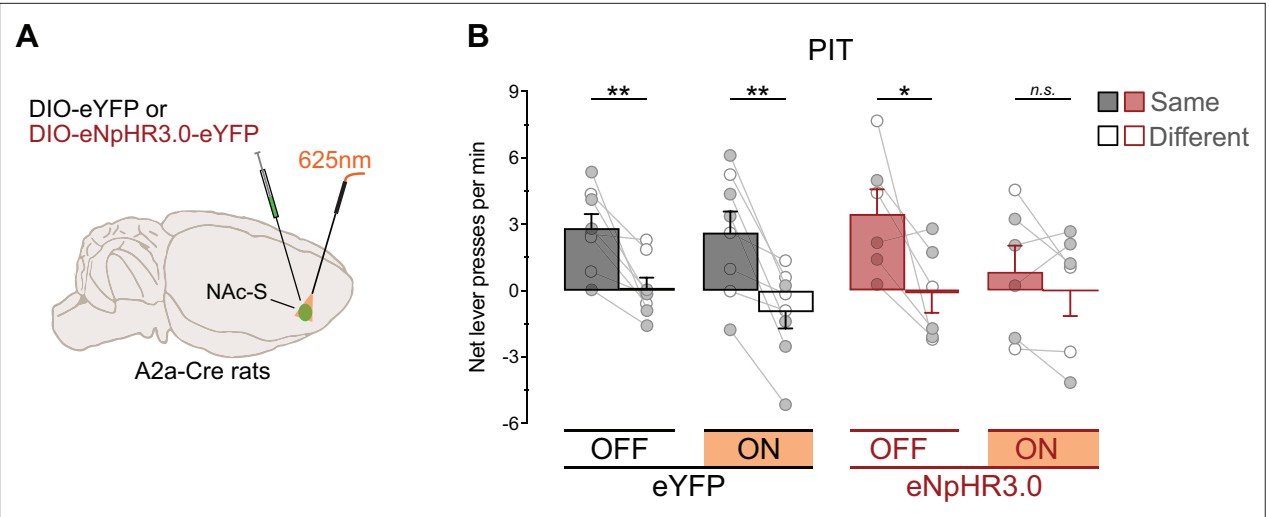

**Figure 4.** NAc-S D2-SPNs mediate for outcome-specific Pavlovian instrumental transfer (PIT). (**A**) A2a-Cre rats were bilaterally infused in the NAc-S with DIO-eYFP (black; 4 females and 4 males) or DIO-eNpHR3.0 (red; 4 females and 2 males). Fiber-optic cannulas were implanted above the NAc-S to provide 625 nm LED illumination (continuous). (**B**) Outcome-specific PIT test: net lever presses when the stimuli predicted the same outcome as the action (Same) or when the stimuli predicted the different outcome (Different). Lever presses are shown for each group in trials conducted under 625 nm LED illumination (ON) and in trials without illumination (OFF). Data are shown as mean ± SEM. Panel B includes individual data points for female (filled circle) and male (open circle) rats. Asterisks denote significant effect (*p < 0.05; **p < 0.01; *n.s.*, nonsignificant).

The online version of this article includes the following figure supplement(s) for figure 4:

**Figure supplement 1.** Histological and behavioral data related to *Figure 4*.

## NAc-S D1-SPNs projections to the VP mediate outcome-specific PIT

Previous work indicates that communication between the NAc-S and VP is critical for outcome-specific PIT (*Leung and Balleine, 2013*). Consistent with the literature (*Lu et al., 1997*; *Kupchik et al., 2015*), we observed that both NAc-S D1- and D2-SPNs densely innervate the VP (*Figure 1E, F*). Since we found that activity in both neuron populations mediates the outcome-specific PIT effect, we investigated whether projections from each population contribute to the effect. We first focused on those originating from NAc-S D1-SPNs. D1-Cre rats were bilaterally infused in the NAc-S with either the null Cre-dependent eYFP virus (eYFP: 3 females and 5 males) or the Cre-dependent halorhodopsin (eNpHR3.0: 6 females and 6 males) virus and were implanted with fiber-optic cannulas above the VP (*Figure 5A*, *Figure 5—figure supplement 1A, B*). The rats then received the behavioral protocol previously described (*Figure 3B*).

Pavlovian and instrumental conditioning went as expected (*Figure 5—figure supplement 1C, D*). The data from the outcome-specific PIT test are presented in *Figure 5B* in the manner described previously since baseline responding did not differ between groups (Group: p = 0.34; see also *Figure 5—figure supplement 1E, F*). Silencing NAc-S D1-SPNs projections to the VP eliminated outcome-specific PIT. Overall lever press rates during the test were similar between groups (Group: p = 0.11). LED activation reduced these rates (LED: $F_{(1,18)}$ = 7.58; $\eta^2$ = 0.30; p < 0.05) but this reduction depended on the group considered (Group x LED: $F_{(1,18)}$ = 9.78; $\eta^2$ = 0.35; p < 0.01). Lever press rates were higher on the action earning the same outcome as the stimuli compared to the action earning the different outcome (Lever: $F_{(1,18)}$ = 35.73; $\eta^2$ = 0.67; p < 0.001), regardless of group (Group x Lever: p = 0.19). There was a Lever by LED light condition interaction (Lever x LED: $F_{(1,18)}$ = 7.56; $\eta^2$ = 0.29; p < 0.05) and critically, there was an interaction between Group, LED light condition, and Lever during the presentation of the predictive stimuli (Group LED x Lever: $F_{(1,18)}$ = 5.01; $\eta^2$ = 0.22; p < 0.05). Follow-up analyses revealed that control eYFP rats expressed outcome-specific PIT whether the LED light was OFF ($F_{(1,7)}$ = 26.63; $\eta^2$ = 0.79; p < 0.001) or ON ($F_{(1,7)}$ = 12.31; $\eta^2$ = 0.64; p < 0.01). By contrast, rats in the eNpHR3.0 group displayed outcome-specific PIT when the light was OFF ($F_{(1,11)}$ = 26.53; $\eta^2$ = 0.71; p < 0.001) but not ON (p = 0.48). Thus, these results show for the first time that NAc-S D1-SPNs mediate outcome-specific PIT via their projections to the VP. Importantly, we also found that the impairment produced by silencing NAc-S D1-SPNs terminals in the VP was restricted to the influence of predictive

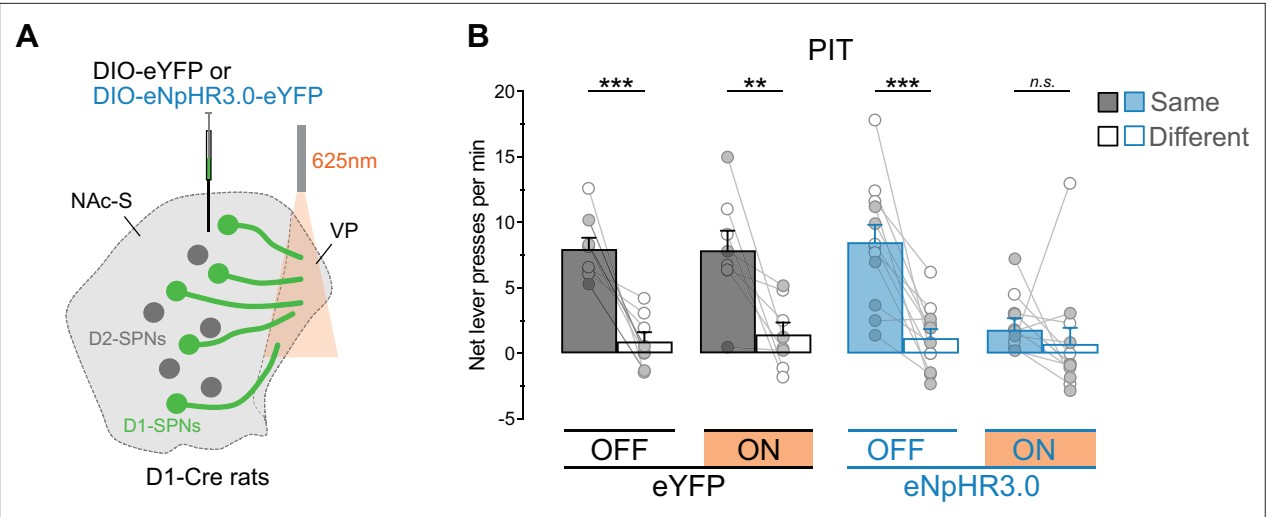

**Figure 5.** NAc-S D1-SPNs projections to the ventral pallidum (VP) mediate outcome-specific Pavlovian instrumental transfer (PIT). (**A**) D1-Cre rats were bilaterally infused in the NAc-S with DIO-eYFP (black; 3 females and 5 males) or DIO-eNpHR3.0 (blue; 6 females and 6 males). Fiber-optic cannulas were implanted above the VP to provide 625 nm LED illumination (continuous). (**B**) Outcome-specific PIT test: net lever presses when the stimuli predicted the same outcome as the action (Same) or when the stimuli predicted the different outcome (Different). Lever presses are shown for each group in trials conducted under 625 nm LED illumination (ON) and in trials without illumination (OFF). Data are shown as mean ± SEM. Panel B includes individual data points for female (filled circle) and male (open circle) rats. Asterisks denote significant effect (**p < 0.01; ***p < 0.01; *n.s.*, nonsignificant).

The online version of this article includes the following figure supplement(s) for figure 5:

**Figure supplement 1.** Histological and behavioral data related to *Figure 5*.

stimuli on choice between actions. In the same rats, this silencing had no effect on value-based choice (*Figure 5—figure supplement 1G*).

## NAc-S D2-SPNs projections to the VP do not mediate outcome-specific PIT

Since activity in NAc-S D2-SPNs is required for outcome-specific PIT (*Figure 3*) and these neurons innervate the VP (*Figure 1F*), we assessed whether D2-SPNs projections to the VP are involved in outcome-specific PIT. A2a-Cre rats were bilaterally infused in the NAc-S with either the null Cre-dependent eYFP virus (eYFP: 3 females and 5 males) or the Cre-dependent halorhodopsin (eNpHR3.0: 2 females and 5 males) virus and were implanted with fiber-optic cannulas above the VP (*Figure 6A*, *Figure 6—figure supplement 1A, B*). The rats then received the behavioral protocol previously described (*Figure 3B*).

Pavlovian and instrumental conditioning went as expected (*Figure 6—figure supplement 1C, D*). The data from the outcome-specific PIT test are presented in *Figure 6B* as previously since baseline responding did not differ between groups (Group: *P*=0.90; see also *Figure 6—figure supplement 1E, F*). Silencing NAc-S D2-SPNs projections to the VP had no effect on outcome-specific PIT. Overall lever press rates were similar between groups (Group: p = 0.56), LED light condition (LED: p = 0.75), and the two factors did not interact (Group x LED: p = 0.73). The rates were higher on the action earning the same outcome as the stimuli relative to the action earning the different outcome (Lever: $F_{(1,13)}$ = 22.71; $\eta^2$ = 0.64; p < 0.001), irrespective of group (Group x Lever: p = 0.55). There was no interaction between Group, LED light condition, and Lever (Group x Light x Lever: p = 0.59). Thus, NAc-S D2-SPNs do not appear to mediate outcome-specific PIT via their projections to the VP. Likewise, we found no evidence that these projections influence value-based choice (*Figure 6—figure supplement 1G*).

## Discussion

The present experiments investigated the role of NAc-S D1- and D2-SPNs in choice between actions using an outcome-specific PIT task. First, they combined anatomical tract-tracing and ex

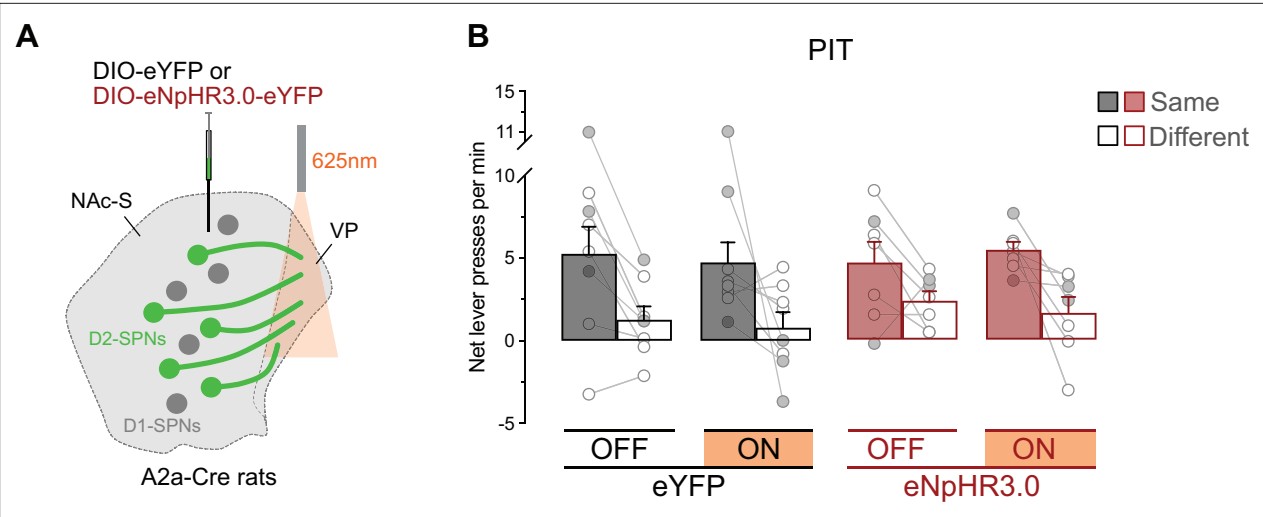

**Figure 6.** NAc-S D2-SPNs projections to the ventral pallidum (VP) do not mediate outcome-specific Pavlovian instrumental transfer (PIT). (**A**) A2a-Cre rats were bilaterally infused in the NAc-S with DIO-eYFP (black; 3 females and 5 males) or DIO-eNpHR3.0 (red; 2 females and 5 males). Fiber-optic cannulas were implanted above the VP to provide 625 nm LED illumination (continuous). (**B**) Outcome-specific PIT test: net lever presses when the stimuli predicted the same outcome as the action (Same) or when the stimuli predicted the different outcome (Different). Lever presses are shown for each group in trials conducted under 625 nm LED illumination (ON) and in trials without illumination (OFF). Data are shown as mean ± SEM. Panel B includes individual data points for female (filled circle) and male (open circle) rats.

The online version of this article includes the following figure supplement(s) for figure 6:

**Figure supplement 1.** Histological and behavioral data related to *Figure 6*.

vivo electrophysiology to demonstrate that two recently developed knock-in rat lines (***Pettibone et al., 2019***) enable selective silencing of activity in either population of NAc-S SPNs. Consistent with previous findings (***Laurent et al., 2014***), they found that NAc-S D1-SPNs are necessary for PIT since their silencing eliminated outcome-specific choice. Additionally, they showed that this choice is also abolished by silencing NAc-S D2-SPNs, providing the first evidence that both SPN populations contribute to PIT expression. Finally, the last two experiments revealed for the first time that outcome-specific choice is likely to involve downstream regulation of VP function by NAc-S D1-SPNs but not NAc-S D2-SPNs. Together, these findings offer novel insights into the cellular mechanisms that govern the outcome-specific influence of predictive stimuli on choice between actions.

Convincing evidence had been provided for a significant role of NAc-S D1-SPNs in PIT (***Laurent et al., 2014***). The evidence was based on the observations that PIT is associated with an increase in ERK1/2 phosphorylation within these SPNs and that pharmacological blockade of NAc-S D1Rs eliminates outcome-specific choice between actions. These findings align with the present study, showing that PIT is eliminated when NAc-S D1-SPNs are silenced during presentations of the predictive stimuli at the time of choice. Importantly, the effect of NAc-S D1-SPNs silencing was specific to the assessment of choice between actions in the presence of predictive stimuli. For instance, the ability of stimuli to elicit approach behaviors toward the magazine remained intact despite NAc-S D1-SPNs silencing, indicating that the silencing effect was not mediated by modulating potential competition between Pavlovian and instrumental responses (***Lovibond, 1981***; ***Holmes et al., 2010***). Furthermore, NAc-S D1-SPNs silencing preserved the capacity to select between actions based on the value of their respective outcomes. This selectivity in the impairment produced by NAc-S D1-SPNs is consistent with previous findings demonstrating that the NAc-S does not contribute to learning and retrieving the S–O and A–O associations produced by Pavlovian and instrumental conditioning (***Corbit and Balleine, 2011***; ***Corbit et al., 2001***; ***Morse et al., 2020***; ***Laurent et al., 2012***; ***Laurent et al., 2014***).

Previous research failed to provide any evidence supporting the involvement of NAc-S D2-SPNs in outcome-specific choice (***Laurent et al., 2014***). Specifically, PIT was found to produce no change in ERK1/2 phosphorylation in these neurons and was left intact by NAc-S D2Rs blockade. However, these assessments have significant limitations, including the widespread distribution of striatal D2Rs (***Gerfen and Surmeier, 2011***) and the inability of ERK1/2 phosphorylation or D2Rs antagonism to capture or

distort activity in NAc-S D2-SPNs (*Bertran-Gonzalez et al., 2008*; *Tozzi et al., 2007*). Further, pharmacological blockade of D2Rs would be expected to enhance D2-SPNs activity, since D2Rs are Gi coupled and so their activation reduces adenylyl cyclase activity. The present study addressed these limitations by implementing optogenetic silencing in a knock-in rat line giving access to D2-SPNs through the targeting of the A2a receptor that is predominantly expressed in these neurons in the NAc-S (*Schiffmann et al., 2007*). We found that PIT is eliminated when NAc-S D2-SPNs are silenced during presentations of the predictive stimuli at the time of choice. Moreover, the same silencing left intact the ability of the predictive stimuli to elicit magazine approaches and preserved the capacity to choose between actions based on the value of their outcomes. Thus, we conclude that both NAc-S D1- and D2-SPNs are indispensable to outcome-specific PIT. Nevertheless, it will be important for future studies to confirm NAc-S D2-SPNs involvement in PIT using alternative approaches, such as chemogenetic or immunohistochemical assessments employing a marker capable of capturing the function of this neuronal population (*Matamales et al., 2020*).

Consistent with the literature (*Lu et al., 1997*; *Kupchik et al., 2015*), we found that both NAc-S D1- and D2-SPNs send dense projections to the VP, which is often described as the major efferent of the nucleus accumbens (*Heimer et al., 1991*; *Zahm and Heimer, 1990*). Importantly, the VP has been shown to play a crucial role in outcome-specific choice (*Leung and Balleine, 2013*; *Leung and Balleine, 2015*; *Leung et al., 2024a*). VP neurons exhibit enhanced cFos activation following an outcome-specific PIT test, and their levels of activation correlate with PIT performance. Moreover, pharmacological inactivation of the VP eliminates outcome-specific choice, and the same elimination is observed when the NAc-S is disconnected from the VP. We therefore tested for a role of NAc-S D1- and D2-SPNs projections to the VP during choice between actions in our outcome-specific PIT task. We found that PIT was abolished by silencing NAc-S D1-SPNs terminals but not by silencing NAc-S D2-SPNs terminals. Neither silencing affected magazine approaches elicited by the predictive stimuli nor choice between actions based on the value of their outcomes. It is important to note that our study does not provide any evidence about the efficacy of NAc-S D2-SPNs terminals silencing in the VP, and future experiments should aim to provide such evidence or adopt other methods to study this pathway. This could involve using opsins with enhanced axonal silencing efficacy (*Mahn et al., 2021*; *Copits et al., 2021*), or employing alternative methods known to disrupt neurotransmitter release such as chemogenetics (*Rost et al., 2022*). Yet, the same silencing for NAc-S D1-SPNs terminals resulted in PIT elimination. Therefore, it seems reasonable to conclude that NAc-S D1-SPNs, but not NAc-S D2-SPNs, projections to the VP are indispensable to observe outcome-specific choice in a PIT task. Since NAc-S D2-SPNs appear to exclusively project to the VP (*Humphries and Prescott, 2010*; *Kupchik et al., 2015*), our findings suggest that these SPNs mediate outcome-specific PIT by locally regulating NAc-S function. By contrast, it remains to be determined whether NAc-S D1-SPNs also coordinate outcome-specific PIT via their projections to the lateral hypothalamus (*O'Connor et al., 2015*) and/or the ventral tegmental area (VTA) (*Humphries and Prescott, 2010*), in addition to the VP.

The present findings are consistent with a recent model proposing that outcome-specific choice in PIT relies on an opioid-based memory residing in the NAc-S (for full description and illustration of the model see *Leung et al., 2024b*; *Laurent and Balleine, 2021*; *Morse et al., 2020*). In this model, as the basolateral amygdala encodes and stores outcome-specific S–O associations across Pavlovian conditioning, it also drives the formation of the NAc-S memory, which involves the durable accumulation of delta-opioid receptors (DOPRs) on the somatic membrane of local CINs (*Bertran-Gonzalez et al., 2013*; *Morse et al., 2020*). Although this memory is not necessary for Pavlovian conditioning per se, its expression is later required to enable the predictive stimuli to guide outcome-specific choice during the PIT test (*Morse et al., 2020*). The model specifically proposes that memory expression is controlled by fluctuations in glutamatergic release from cortical inputs and dopamine release from projections originating from the VTA, as this brain region has been found to be important for outcome-specific PIT (*Corbit et al., 2007*; *Sias et al., 2024*; *Seitz et al., 2022*; *Leung and Balleine, 2015*). One consequence is to activate NAc-S D2-SPNs and enkephalin discharge by these neurons. Enkephalin then binds onto DOPRs that had accumulated on local CINs and dampens acetylcholine secretion from the interneurons. The sudden drop in NAc-S acetylcholine frees D1-SPNs from the inhibitory tone imposed by acetylcholine occupancy of muscarinic M4 receptors that are exclusively found on these SPNs (*Tayebati et al., 2004*; *Lobo et al., 2006*; *Guo et al., 2010*; *Jeon et al., 2010*). The ultimate consequence is to promote NAc-S D1-SPNs function, including the coordination of

outcome-specific choice in PIT by regulating activity in downstream brain regions such as the VP. The model predicts all the findings presented here. NAc-S D2-SPNs silencing eliminates PIT at the time of choice by preventing enkephalin release and thereby expression of the DOPR-based memory. The main consequence of this expression is precluded by NAc-S D1-SPNs silencing at the time of choice, while silencing the terminals of these SPNs in the VP squashes their ability to regulate the activity of this brain region to coordinate outcome-specific choice in PIT. Thus, one fundamental implication of the present findings is to strengthen the proposal that an opioid-based memory system in the NAc-S enables outcome-specific choice between actions.

In summary, the present experiments found that the two main populations of SPNs in the NAc-S are indispensable for outcome-specific choice between actions in a PIT task. They also revealed that NAc-S D1-SPNs coordinate this choice by downstream regulation of VP activity. By contrast, NAc-S D2-SPNs function in PIT appears to be restricted to modulating local activity. These findings provide novel insights into the cellular mechanisms and circuitry underlying the outcome-specific influence of predictive stimuli on choice between actions and are consistent with a recent model proposing that this influence is mediated by an opioid-based memory system in the NAc-S. Beyond these mechanistic considerations, the findings offer an opportunity to gain novel insights about various disorders in which the outcome-specific influence of predictive stimuli over our choices and decisions is dysfunctional. These disorders include depression (*Geurts et al., 2013*; *Nord et al., 2018*), anxiety disorders (*Quail et al., 2017*; *Krypotos and Engelhard, 2020*), substance-use disorders (*Heinz et al., 2019*; *Hogarth et al., 2019*; *Hogarth et al., 2019*; *Steins-Loeber et al., 2020*; *Garbusow et al., 2016*; *Garbusow et al., 2019*; *Hogarth et al., 2019*), gambling disorders (*Genauck et al., 2019*), anorexia nervosa (*Vogel et al., 2020*), and obesity (*Watson et al., 2014*; *Lehner et al., 2017*; *Meemken and Horstmann, 2019*).

## Materials and methods
### Subjects

135 rats from two genetically modified knock-in lines were used and obtained from the breeding facility at the University of New South Wales (Sydney, Australia). D1-Cre rats (Rat Research & Resource Center, Columbia, MO, USA; LE-Drd1$^{em1(iCre)Berke}$, RRRC#: 00856) expressed Cre recombinase in neurons expressing the dopamine D1 receptors (D1R). A2a-Cre rats (Rat Research & Resource Center, Columbia, MO, USA; LE-*Adora2a*$^{em1(iCre)Berke}$, RRRC#: 00857) expressed Cre recombinase in neurons expressing the adenosine A2A receptors (A2R). All rats were heterozygous and generated by crossing a heterozygous male with a Long-Evans wild-type female obtained from a colony maintained by the University of New South Wales (founding animals were sourced from Envigo/Inotiv, Blue Spruce outbred; HsdBlu:LE). Genotyping was completed by sending ear clippings to Transnetyx (Cordova, TN, USA). The rats were at least 8 weeks old at the beginning of each experiment. Efforts were made to allocate an equal number of female and male rats in each group. However, these efforts were hampered by exclusions following post-mortem assessments of viral spread and fiber-optic cannula placement. The final number of female and male rats did not provide sufficient power to analyze an effect of sex (our previous work found no influence of sex on PIT or value-based choice; *Burton et al., 2024*). Therefore, all test data present individual performance for female and male rats. Rats were housed in transparent plastic boxes with their littermates (up to four rats per box) throughout, and in a climate-controlled colony room maintained on a 12-hr light–dark cycle (lights on at 7:00 am). Behavioral procedures were conducted during the light phase (8:00 am to 6:00 pm). Water and standard lab chow were available ad libitum prior to the start of each experiment. Rats were food restricted to maintain them at ~90% of their ad libitum body weight during the behavioral protocols. Food restriction was initiated 3–5 days prior to the start of the protocols and was maintained throughout all training and testing phases. Rats received a daily food amount, which was adjusted based on body weight measurements recorded every 2 days. This study was performed in strict accordance with the recommendations in the Guide for the Care and Use of Laboratory Animals of the National Health and Medical Research Council in Australia. All of the animals were handled according to approved Animal Care and Ethics Committee (ACEC) protocols of the University of New South Wales. The experimental protocols were approved by the UNSW ACEC (Permit Number: 20/35A). All surgery was performed under isoflurane anesthesia, and every effort was made to minimize suffering.

## Experimental timeline

Rats underwent initial stereotaxic surgery for viral infusion, followed by 7–11 days recovery. After recovery, rats received 8 days of Pavlovian conditioning and 8 days of instrumental conditioning, except in the VP experiments (NAc-S D1- and D2-SPNs → VP projection experiments) where only 7 days of instrumental conditioning were given. Following a 3- to 4-day interval, rats received a second surgery to implant fiber-optic cannulas and 3–4 days additional recovery. Rats then completed reminder sessions: one Pavlovian and two instrumental conditioning sessions, except in the VP experiments (NAc-S D1-SPNs and D2-SPNs → VP projection experiments) where three instrumental sessions were given. Testing began the following day with the PIT test, followed by two instrumental conditioning reminder sessions over the next 2 days. Finally, rats underwent outcome devaluation using sensory-specific satiety and choice tests over two consecutive days.

## Viruses

The following viruses were used: AAV5-EF1a-DIO-eYFP (eYFP; AddGene; Watertown, MA, USA) and AAV5-EF1a-DIO-eNpHR3.0-eYFP (eNpHR3.0; UNC Vector Core; Chapel Hill, NC, USA).

## Surgeries

Stereotaxic surgery was conducted under isoflurane gas anesthesia (0.8 l/min; induction at 4–5%, maintenance 2–2.5%). Animals were placed in a stereotaxic frame (Kopf Instruments, Tujunga, CA, USA), the surgical area was shaved, and betadine iodine antiseptic solution was applied. At the incision site, bupivacaine hydrochloride (0.5%; 0.1–0.2 ml) was injected subcutaneously. In the lower flank, Metacam (1 mg/kg) was injected subcutaneously. An incision was made along the midline to expose the skull, the membrane on top of the skull was cleared, and the skull was adjusted to align bregma and lambda on the same horizontal plane.

For viral infusion, small holes were drilled in the skull above the DS or NAc-S. The following coordinates (indicated in mm from bregma) were used for the DS: –0.45 anteroposterior (A/P); ±2.6 (M/L); –4.85 dorsoventral (D/V). For the NAc-S, the coordinates were: +1.8 or +1.9 A/P; ±0.85 or ±0.95 mediolateral M/L; –7.7 or –7.85 D/V. An Infuse/Withdraw Pump (Standard Infuse/Withdraw Pump 11 Elite; Holliston, Harvard Apparatus, MA, USA) in combination with glass 1 µl syringes (86200; Hamilton Company, Reno, NV, USA) was used to perform infusions. 0.5 µl of the virus was infused in each hemisphere at a rate of 0.1 µl/min. Following each infusion, the needle was left in position for 5–10 min to allow for diffusion. The incision site was closed and secured with surgical sutures (684G; Ethicon, North Ryde, NSW, Australia) or surgical staples (59027; Stoelting, Wood Dale, IL, USA), and betadine antiseptic ointment was applied to the area. Animals then received subcutaneous injections of the antibiotic Duplocillin (0.15 ml/kg) and sodium chloride (0.9%, 2 ml) and placed on a heat mat for recovery. Once alert and responsive, animals were returned to their home cages and monitored daily as they recovered for a minimum of 1 week following surgery with ad libitum access to food and water.

Rats received a second surgery to bilaterally implant fiber-optic cannulas (10 mm length, 0.48 mm diameter; Doric Lenses, Québec, QC, Canada) after completion of the initial 15–16 days of training (Pavlovian and Instrumental). The surgical area was prepared using the same method as described for viral surgeries. Three jewelers' screws were inserted into the skull, distributed anterior and posterior to the implantation site and secured in place with dental cement. Fiber-optic cannulas were implanted into the targeted brain region. The following coordinates (indicated in mm from bregma) were used for the NAc-S: +1.8 or +1.9 A/P; –0.85 or –0.95 M/L for left hemisphere; +2.2 M/L with –10° angle for right hemisphere; –6.75 D/V for left hemisphere; –6.8 or –6.9 D/V for right hemisphere. The following coordinates (indicated in mm from bregma) were used for the VP: +0.24 A/P; ±1.8 M/L; –7.5 or –8.0 D/V. Post-implantation, rats received the same care as described for viral surgeries and were allowed to recover for 3–4 days before resuming the behavioral procedures.

## Behavioral apparatus

Training and testing were conducted in 12 identical operant chambers (ENV-007-VP; L 29.53 x W 23.5 x H 27.31 cm; MED Associates, Fairfax, VT, USA) enclosed within light- and sound-resistant cabinets (ENV-018MD; L 59.69 x W 40.64 x H 55.88 cm; MED Associates, Fairfax, VT, USA). The chambers consisted of two stainless-steel side walls, three clear polycarbonate walls and ceiling, and a stainless-steel grid floor. Each chamber was equipped with a recessed magazine connected to a 45-mg grain

pellet (F0165; Bio-Serv Biotechnologies, Flemington, NJ, USA; one pellet per delivery) dispenser and a pump fitted with a 60-ml syringe that delivered a 0.2-ml sucrose solution (20% wt/vol; delivered across 2 s; white sugar; Winc, Erskine Park, NSW, Australia) into the left side of the magazine. Head entries into the magazine were detected via an infrared beam that crossed the magazine opening. One retractable lever was located on either side of the magazine. A house light (3 W, 24 V) situated at the top of the wall opposite to the food magazine was used to illuminate the chamber during the training and test sessions. Each chamber contained a Sonalert white noise generator (80 dB), and a 28 V DC mechanical relay that delivered a 2-Hz clicker stimulus. Training and testing sessions were programmed and controlled by computers external to the testing rooms using MED-PC V software (MED Associates, Fairfax, VT, USA) which also recorded experimental data from each session. All cabinets were fitted with cameras (IPC-K35AP; Dahua, Artarmon, NSW, Australia) to view real-time chamber activity using D-ViewCam DCS-100 software (D-Link, North Ryde, NSW, Australia). The ceiling of these chambers contained a hole which enabled the connection of the patch cable between the LED and each freely moving rat.

Outcome devaluation was conducted in a separate room equipped with 16 individual, open-top plastic boxes with stainless steel wire mesh lids. During devaluation, the lights were dimmed, and each individual chamber was fitted with either a ceramic dish for grain pellet devaluation or a plastic sipper bottle for sucrose solution devaluation.

## Optogenetic equipment

Fiber-optic cannulas (10 mm; Doric Lenses, Québec, QC, Canada) consisted of a zirconia ferrule contained within a stainless-steel receptacle from which a silica/polymer flat-tip fiber extended from. Fiber-optic patch cords (custom; Doric Lenses, Québec, QC, Canada) were enclosed in a flexible cladding that consisted of one transparent fiber that split into two smaller fibers for attachment to the implanted fiber-optic cannulas. Delivery of the LED light was controlled through a TTL adapter which converted 28 V DC output to a TTL transition (MED Associates, Fairfax, VT, USA). The adapter was connected to an LED driver (Doric Lenses, Québec, QC, Canada) and Connectorized LED light source (Doric Lenses, Québec, QC, Canada) with a fiber-optic rotary joint for attachment to the fiber-optic patch cords. LED light was delivered as orange light (625 nm; continuous) and measured at least 10 mW at the cannula tip using a photometer (PM200; ThorLabs, Newton, NJ, USA) when the LED was switched on.

## Behavioral procedures

### Pavlovian conditioning

Pavlovian conditioning involved 8 daily 1 hr training sessions. Two auditory conditioned stimuli (S1 and S2; white noise or clicker) were paired with two food outcomes (O1 and O2; grain pellets or sucrose solution). Each stimulus was presented for 2 min in duration, four times each in a pseudo-random order within each training session. Stimulus presentations were separated by pseudorandom intertrial intervals (ITIs) ranging between 3 and 5 min in length (4 min average). The stimulus order and ITIs were changed daily. S–O pairings were counterbalanced within- and between-groups. Food outcomes were delivered on a random time 30 s schedule throughout the stimulus presentation. Magazine entries were recorded throughout the session, and conditioned responding was analyzed by comparing magazine entries in the 2 min prior to stimulus presentation ('pre S') and during the 2 min stimulus presentation period.

### Instrumental conditioning

During instrumental conditioning, two actions (A1 and A2; left or right lever presses) were trained with the two different outcomes (O1 and O2) in 2 separate daily sessions for 8 days. The order of the sessions was counterbalanced, as were the A–O relationships, which were further counterbalanced with the S–O relationships. Each session ended when 20 outcomes were earned or when 30 min had elapsed. For the first 2 days, actions were continuously reinforced (i.e., one action earned one outcome delivery). Across the next three sessions, actions were rewarded on a random ratio 5 (RR5) schedule (i.e., each action delivered an outcome with a probability of 0.2). For the final 3 days, animals were rewarded on an RR10 schedule (i.e., each response delivered an outcome with a probability of 0.1). In the VP experiments (NAc-S D1- and D2-SPNs → VP projections), rats only received 2 days on

RR10. Lever press actions were measured throughout the duration of the session, and performance during instrumental training was assessed using the mean number of lever presses per min.

## Pavlovian instrumental transfer test

Across the 3 days preceding this test, rats were given one session of Pavlovian retraining and two separate sessions of instrumental retraining. In the VP experiments (NAc-S D1- and D2-SPNs → VP projections), rats first received 1 day of Pavlovian conditioning followed by 3 days of instrumental conditioning (RR10). Immediately before the test, patch cables were attached to the fiber-optic cannulas and the LED. During the test, both levers were inserted into the chamber simultaneously and remained in the chamber for the duration of the test. The test was conducted in extinction (i.e., no outcomes were delivered). To reduce baseline (pre-S) lever presses, responding on both levers was extinguished for 8 min prior to any stimuli being presented. Each stimulus was presented for 2 min, four times each during the test in the following order: noise-clicker-clicker-noise-clicker-noise-noise-clicker. Each stimulus presentation was separated by a fixed 3-min interval. Critically, the LED was activated during half of the trials (ON) but left inactive during the other half (OFF). The order for ON and OFF trials was counterbalanced. Responses were recorded prior to and during the S presentation. These were then collated into 'Baseline' (lever press rates in the two min prior to S presentation), 'Same' (when S shares the *same* outcome with the action) and 'Different' (when S has a *different* outcome with the action) following data collection.

## Choice tests following outcome devaluation

Across the two consecutive days preceding these tests, rats were given two daily instrumental conditioning sessions (RR10). Following the second session, rats were habituated to the devaluation chambers for 30 min. During these habituation sessions, animals had access to one outcome (counterbalanced). On the first test day, rats were placed in the devaluation chambers for 1 hr where they had ad libitum access to one of the previously trained outcomes (O1 or O2). Once removed from the devaluation boxes, patch cables were attached to the fiber-optic cannulas and the LED, and the first choice test was conducted. Both levers were made available for 2 min each, four times during the session with an ITI of 2 min where the levers were retracted. The LED was activated during two of the 2 min trials where the levers were available (ON) and was left inactive during the other two trials (OFF). The order of the ON and OFF trials was counterbalanced. A similar procedure was implemented the following day, except that the other outcome was devalued. Lever presses and magazine entries were recorded while the levers were available.

## Tissue preparation and histology

### Transcardial fixation and brain sectioning

Rats were euthanized with an intraperitoneal injection of sodium pentobarbital (1 ml) and transcardially perfused with 4% paraformaldehyde (PFA) in 0.1 M sodium phosphate buffer (PBS, pH 7.5; 400 ml for rats). Brains were immediately removed and placed in individual specimen jars containing PFA and stored at 4°C overnight for post-fixation. Brains were sliced into 40 µm coronal or sagittal sections in 0.1 M Tris-buffered saline (TBS, 0.25 M Tris, 0.5 M NaCl, pH 7.5) using a vibratome (VT1200; Leica Microsystems, North Ryde, NSW, Australia) and stored at –20°C in cryoprotective solution containing 30% ethylene glycol, 30% glycerol and 0.1 M PBS.

### Immunofluorescence

Free-floating sections were selected in TBS to contain the region of interest (ROI) and washed three times for 10 min each in TBS solution. Sections were then incubated in 0.5% Triton-X, 10% Normal Horse Serum (NHS) in TBS blocking solution for 2 hr. Sections were washed again in TBS solution three times for 10 min each. Sections were then incubated between 12 and 48 hr (dependent on antibody) in a solution containing the primary antibodies in 0.2% Triton-X, 2% NHS in TBS at 4°C. The primary antibodies were the following: purified mouse anti-DARPP-32 (1:300; BD Biosciences, Macquarie Park, Australia; #AB398980) and polyclonal rabbit anti-GFP (1:1000; Invitrogen, Mulgrave, Australia; #AB221569). After the specified incubation period, the sections were washed three times in TBS for 10 min each and then transferred into a solution containing the secondary antibodies in TBS solution

at room temperature for 1 hr. The secondary antibodies were the following: donkey anti-mouse Cy3 (1:400; Jackson ImmunoResearch Laboratories, West Grove, USA; #AB2340813) and donkey anti-rabbit Alexa-488 (1:500; Jackson ImmunoResearch Laboratories, West Grove, USA; #AB2313584). Next, sections were washed three times in TBS for 10 min and then mounted onto glass slides using Vectashield hardset antifade mounting medium (H-1400; Vector Laboratories, Newark, CA, USA). Sections that did not require immunofluorescence staining were washed three times for 10 min each in TBS and mounted using Vectashield hardset antifade mounting medium.

## Microscopy and imaging

Imaging of the mounted brain sections was performed on a spinning disk confocal microscope (Zyla 4.2 sCMOS; Andor Technology, Belfast, UK) using the Diskovery multi-modal imaging platform (Oxford Instruments, Belfast, UK). The Diskovery platform was mounted on an inverted microscope body with a motorized stage and Perfect Focus System (Eclipse Ti; Nikon, Melville, NY, USA). Image capture was controlled using the NIS-Elements Software (Nikon, Melville, NY, USA). Single high-magnification field-of-view images were generated by automatic stitching of multiple adjacent frames from within a defined area using the motorized stage (optical magnification: 20×; pixel depth: 16-bit). All images were processed using Fiji ImageJ open-source software (*Rueden et al., 2017*). Additional images were taken with a confocal microscope (BX61W1; Olympus, Shinjuku, Tokyo, Japan) using Fluoview software (FV1000; Olympus, Shinjuku, Tokyo, Japan) for further placement analysis. The images were used to assess neuronal pathways, as well as the spread of viral infusions, placement of fiber-optic cannulas, and potential damages due to the surgical procedures. Based on the latter, the following numbers of rats were excluded from the final statistical analyses due to misplaced viral or fiber-optics misplacements: NAc-S D1-SPNs silencing ($n$ = 9 females and 9 males), NAc-S D2-SPNs silencing ($n$ = 4 females and 5 males), NAc-S D1-SPNs → VP silencing ($n$ = 5 females and 10 males), NAc-S D2-SPNs → VP silencing ($n$ = 4 females and 1 male).

### Immunofluorescence quantifications

The specificity of viral transduction in D1-Cre and A2-Cre rats (*Figure 1*) was achieved through manual cell counting and involved calculating a percentage of the total number of fluorophore-labeled (eYFP) cells relative to the total number of DARPP-32-positive cells, and a percentage of the total number of DARPP-32-positive cells relative to the total number of fluorophore-labeled (eYFP) cells. Two slices per rat were selected around the following anteroposterior coordinates (in mm relative to bregma): –0.45 for DS and +1.8 for NAc-S. Before quantification, all images were randomly renumbered, and cell counts were performed in Fiji ImageJ in one ROI per image (ROI size: 0.25 mm$^2$ for both DS and NAc-S). The ROI was defined in the eYFP channel around the viral injection site. The average number (± SD) of DARPP-32-positive cells per ROI was: 194 ± 59 for DS and 132 ± 24 for NAc-S.

## Electrophysiological recordings

### Brain slice preparation

Rats were euthanized under isoflurane gas anesthesia (4% in air). Brains were immediately removed and sliced using a vibratome (VT1200; Leica Microsystems, North Ryde, NSW, Australia) in ice-cold oxygenated sucrose buffer solution (consisting of [in mM]: 241 sucrose, 28 NaHCO$_3$, 11 glucose, 1.4 NaH$_2$PO$_4$, 3.3 KCl, 0.2 CaCl$_2$, 7 MgCl$_2$). Coronal brain slices (300 μm thickness) containing the NAc were sampled and maintained at 33°C in a submerged chamber containing physiological saline (consisting of [in mM]: 124 NaCl, 2.5 KCl, 1.25 NaH$_2$PO$_4$, 1 MgCl$_2$, 1 CaCl$_2$, 10 glucose and 26 NaHCO$_3$) and equilibrated with 95% O$_2$ and 5% CO$_2$.

### Recordings

Following 1 hr equilibration, slices were transferred to a recording chamber. An upright micro-scope (BX50WI; Olympus, Shinjuku, Tokyo, Japan) was used to visualize neurons using differential interference contrast (DIC) Dodt tube optics and eYFP fluorescence, and superfused continuously (1.5 ml/min) with oxygenated physiological saline at 33°C. Whole-cell patch-clamp recordings were performed using glass pipettes (tip size 2–5 MΩ) containing internal solution (consisting of [in mM]: 115 K gluconate, 20 NaCl, 1 MgCl$_2$, 10 HEPES, 11 EGTA, 5 Mg-ATP, 0.33 Na-GTP, and 5 phospho-creatine di(tris) salt, pH 7.3, osmolarity 285–290 mOsm/l). Neurons sampled during recording were

marked with Biocytin (0.1%) added to the internal solution. A Multiclamp 700B amplifier (Molecular Devices, San Jose, CA, USA) connected to a Macintosh computer and interface ITC-18 (InstruTECH, Longmont, CO, USA) was used for data acquisition. Liquid junction potentials of –10 mV were not corrected. In current-clamp mode, membrane potentials were sampled at 5 kHz (low pass filter 2 kHz; Axograph X, Axograph, CA, USA). A stock solution of drug was diluted to working concentration in the extracellular solution immediately before use and applied by continuous superfusion. Data from whole-cell recordings were included in analyses if the neurons appeared healthy under DIC on the monitor screen.

## LED stimulation

Transfected neurons with eNpHR3.0 or eYFP in the NAc-S were illuminated by a continuous wave of LED light (625 nm, 2 min, 2 mW; ThorLabs, Newton, NJ, USA) onto the brain slice under the 40X water-immersion objective. Since SPNs in ex vivo brain slices are not spontaneously active, brief injection of positive current step pulses was applied to make them fire action potentials.

## Post hoc histological analysis

Immediately after physiological recording, brain slices were fixed overnight using 4% PFA in 0.16 M phosphate buffer (PB) solution, washed, and then transferred into 0.5% Triton-X in PB solution for 3 days to permeabilize cells. Slices were washed, mounted onto glass slides, and then coverslipped with Fluoromount-G mounting medium (Southern Biotech, Birmingham, AL, USA). A 2D projection image was then obtained from a collated image stack using the confocal microscope (X61W1; Olympus, Shinjuku, Tokyo, Japan).

## Statistical analyses

The data presented met the assumptions of the statistical test used. The main measures were the number of magazine entries and lever presses per minute. The difference between groups was analyzed using a planned orthogonal contrast procedure controlling the per contrast error rate (*Hays, 1963*). A first contrast (Group) compared rats that had been infused with the control eYFP virus and rats that had been infused with the inhibitory eNpHR3.0 virus. Changes across days (Day) during Pavlovian and instrumental conditioning were assessed by planned linear trend analyses. Pavlovian conditioning and the PIT tests included a within-subject factor (Period) comparing magazine entry rates during the 2-min stimuli against rates during a 2-min baseline period immediately preceding the stimuli. During the PIT tests, a within-subject factor (Lever) examined the net (i.e., minus baseline) rates of lever presses on the response (Same) earning the same outcome as predicted by the stimulus against those on the response (Different) earning the different outcome. A second within-subject factor (LED) examined these rats when the LED was activated (ON) or not activated (OFF). The choice tests following outcome devaluation used a within-subject factor of response identity (Lever) comparing rates of lever presses on the response (Valued) earning the valued outcome against those on the response (Devalued) earning the devalued outcome. It also used the LED within-subject factor described above. All analyses were carried out using the PSY statistical software (School of Psychology, the University of New South Wales, Australia). The Type I error rate was controlled at alpha = 0.05 for each contrast tested. If interactions were detected, follow-up simple effects analyses were calculated to determine the source of the interactions. For each significant statistical comparison, we report measures of effect size partial eta-squared ($\eta^2$; $\eta^2 = 0.01$ is a small effect, $\eta^2=0.06$ a medium effect, and $\eta^2=0.14$ a large effect). The required number of rats per group was determined during the design stage of the study and was based on our prior studies of specific Pavlovian instrumental transfer.

## Acknowledgements

The research reported in this manuscript was supported by a Future Fellowship from the Australian Research Council (FT220100474) to VL.

## Additional information

### Funding

| Funder | Grant reference number | Author |
| --- | --- | --- |
| Australian Research Council | FT220100474 | Vincent Laurent |

The funders had no role in study design, data collection, and interpretation, or the decision to submit the work for publication.

### Author contributions

Octavia Soegyono, Data curation, Formal analysis, Investigation, Methodology, Writing – review and editing; Elise Pepin, Beatrice K Leung, Formal analysis, Investigation, Writing – review and editing; Billy Chieng, Data curation, Investigation, Writing – review and editing; Bernard W Balleine, Resources, Writing – review and editing; Vincent Laurent, Conceptualization, Resources, Data curation, Formal analysis, Supervision, Funding acquisition, Validation, Investigation, Methodology, Writing - original draft, Project administration, Writing – review and editing

### Author ORCIDs

Octavia Soegyono (iD) https://orcid.org/0009-0005-8573-5501
Elise Pepin (iD) https://orcid.org/0009-0001-2707-8081
Beatrice K Leung (iD) https://orcid.org/0000-0002-6471-4526
Bernard W Balleine (iD) https://orcid.org/0000-0001-8618-7950
Vincent Laurent (iD) https://orcid.org/0000-0003-2333-8437

### Ethics

This study was performed in strict accordance with the recommendations in the Guide for the Care and Use of Laboratory Animals of the National Health and Medical Research Council in Australia. All of the animals were handled according to approved Animal Care and Ethics Committee (ACEC) protocols of the University of New South Wales. The experimental protocols were approved by the UNSW ACEC (Permit Number: 20/35A). All surgery was performed under isoflurane anesthesia, and every effort was made to minimize suffering.

Reviewer #1 (Public review): https://doi.org/10.7554/eLife.107566.4.sa1
Reviewer #2 (Public review): https://doi.org/10.7554/eLife.107566.4.sa2
Author response https://doi.org/10.7554/eLife.107566.4.sa3

## Additional files

### Supplementary files

MDAR checklist

### Data availability

All data generated and/or analyzed during this study are available via the Open Science Framework repository (https://osf.io/6yrxc/). The files contain the numerical data used to generate the figures.

The following dataset was generated:

| Author(s) | Year | Dataset title | Dataset URL | Database and Identifier |
| --- | --- | --- | --- | --- |
| Soegyono O, Pepin E, Leung BK, Chieng BC, Balleine BW, Laurent V | 2025 | Source data for eLife article "The influence of nucleus accumbens shell D1 and D2 neurons on outcome-specific Pavlovian instrumental transfer" | https://osf.io/6yrxc/ | Open Science Framework, 6yrxc |

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

## Appendix 1

### Magazine entry rates during the PIT tests

NAc-S D1-SPNs silencing enhanced magazine entry rates during the outcome-specific PIT test (*Figure 3—figure supplement 1F*). During this test, stimuli elicit both magazine entries and lever presses, with these response types competing for behavioral control (*Lovibond, 1981*). Manipulations that reduce one response type might therefore be expected to enhance the other. Since NAc-S D1-SPNs silencing impaired outcome-specific PIT and reduced lever press responding (*Figure 3C*, *Figure 3—figure supplement 1E*), it would seem reasonable to conclude that this reduction resulted in increased magazine entry rates. However, this increase was also observed during LED OFF trials in eNpHR3.0 rats (*Figure 3—figure supplement 1F*), when lever press performance was comparable to control eYFP rats (*Figure 3C*, *Figure 3—figure supplement 1E*). Therefore, response competition cannot explain the enhanced magazine entry rates. The results from NAc-S D2-SPNs silencing further rule out this possibility. While D2-SPNs silencing impaired outcome-specific PIT (*Figure 4C*, *Figure 4—figure supplement 1E*), it had no significant effect on magazine entry rates during PIT, unlike D1-SPNs silencing. If anything, there was a trend toward reduced magazine entries with D2-SPNs silencing, but this reduction was also present during LED OFF conditions. Notably, a similar trend occurred during Pavlovian training, when lever press manipulanda were unavailable and no silencing had been implemented (*Figure 4—figure supplement 1C*). Overall, the present data do not provide a clear explanation for the enhanced magazine entry rates observed in eNpHR3.0 rats during the PIT test. Future studies using pure Pavlovian tasks without a competing instrumental response may be necessary to establish whether NAc-S SPNs modulate magazine entry performance.

## Appendix 2

### Choice tests under NAc-S D1-SPNs silencing

The choice test data presented in *Figure 3—figure supplement 1G* revealed a significant Group × Lever × LED interaction. Follow-up analyses demonstrated that all groups retained the capacity to select between actions based on outcome value. However, the significant interaction suggested that NAc-S D1-SPNs silencing marginally attenuated this capacity. Caution is warranted before interpreting this finding as evidence for a role of NAc-S D1-SPNs in value-based decision-making. Several studies have demonstrated that NAc-S manipulations typically preserve such choice behavior (*Corbit et al., 2001*; *Corbit and Balleine, 2011*; *Laurent et al., 2012*). Furthermore, previous work showed that NAc-S D1 receptor blockade impairs outcome-specific PIT while leaving value-based choice unaffected (*Laurent et al., 2014*). We therefore favor an alternative explanation for the observed marginal reduction. As shown in *Figure 3—figure supplement 1A*, viral transduction extended slightly into the nucleus accumbens core (NAc-C), a region critical for value-based decision-making (*Corbit et al., 2001*; *Corbit and Balleine, 2011*; *Laurent et al., 2012*; *Parkes et al., 2015*). The marginal impairment may therefore reflect inadvertent silencing of a small number of NAc-C D1-SPNs rather than a functional contribution from NAc-S D1-SPNs. Future studies specifically targeting a larger population of NAc-C D1-SPNs populations would help clarify this possibility.

