## [Editor Report · eLife Assessment]

This study provides novel and **convincing** evidence that both dopamine D1 and D2 expressing neurons in the nucleus accumbens shell are crucial for the expression of cue-guided action selection, a core component of decision-making. The research is systematic and rigorous in using optogenetic inhibition of either D1- or D2-expressing medium spiny neurons in the NAc shell to reveal attenuation of sensory-specific Pavlovian-Instrumental transfer, while largely sparing value-based decision on an instrumental task. The **important** findings in this report build on prior research and resolve some conflicts in the literature regarding decision-making.

---

## [Referee Report · Reviewer #1 (Public review)]

In the current article, Octavia Soegyono and colleagues study "The influence of nucleus accumbens shell D1 and D2 neurons on outcome-specific Pavlovian instrumental transfer", building on extensive findings from the same lab. While there is a consensus about the specific involvement of the Shell part of the Nucleus Accumbens (NAc) in specific stimulus-based actions in choice settings (and not in General Pavlovian instrumental transfer - gPIT, as opposed to the Core part of the NAc), mechanisms at the cellular and circuitry levels remain to be explored. In the present work, using sophisticated methods (rat Cre-transgenic lines from both sexes, optogenetics and the well-established behavioral paradigm outcome-specific PIT - sPIT), Octavia Soegyono and colleagues decipher the differential contribution of dopamine receptors D1 and D2 expressing-spiny projection neurons (SPNs).

After validating the viral strategy and the specificity of the targeting (immunochemistry and electrophysiology), the authors demonstrate that while both NAc Shell D1- and D2-SPNs participate in mediating sPIT, NAc Shell D1-SPNs projections to the Ventral Pallidum (VP, previously demonstrated as crucial for sPIT), but not D2-SPNs, mediates sPIT. They also show that these effects were specific to stimulus-based actions, as value-based choices were left intact in all manipulations.

This is a well-designed study and the results are well supported by the experimental evidence. The paper is extremely pleasant to read and add to the current literature.

---

## [Referee Report · Reviewer #2 (Public review)]

Summary:

This manuscript by Soegyono et a. describes a series of experiments designed to probe the involvement of dopamine D1 and D2 neurons within the nucleus accumbens shell in outcome-specific Pavlovian-instrumental transfer (osPIT), a well-controlled assay of cue-guided action selection based on congruent outcome associations. They used an optogenetic approach to phasically silence NAc shell D1 (D1-Cre mice) or D2 (A2a-Cre mice) neurons during a subset of osPIT trials. Both manipulations disrupted cue-guided action selection but had no effects on negative control measures/tasks (concomitant approach behavior, separate valued guided choice task), nor were any osPIT impairments found in reporter only control groups. Separate experiments revealed that selective inhibition of NAc shell D1 but not D2 inputs to ventral pallidum were required for osPIT expression, thereby advancing understanding of the basal ganglia circuitry underpinning this important aspect of decision making.

Strengths:

The combinatorial viral and optogenetic approaches used here were convincingly validated through anatomical tract-tracing and ex vivo electrophysiology. The behavioral assays are sophisticated and well-controlled to parse cue and value guided action selection. The inclusion of reporter only control groups is rigorous and rules out nonspecific effects of the light manipulation. The findings are novel and address a critical question in the literature. Prior work using less decisive methods had implicated NAc shell D1 neurons in osPIT but suggested that D2 neurons may not be involved. The optogenetic manipulations used in the current study provides a more direct test of their involvement and convincingly demonstrate that both populations play an important role. Prior work had also implicated NAc shell connections to ventral pallidum in osPIT, but the current study reveals the selective involvement of D1 but not D2 neurons in this circuit. The authors do a good job of discussing their findings, including their nuanced interpretation that NAc shell D2 neurons may contribute to osPIT through their local regulation of NAc shell microcircuitry.

Weaknesses:

The current study exclusively used an optogenetic approach to probe the function of D1 and D2 NAc shell neurons. Providing a complementary assessment with chemogenetics or other appropriate methods would strengthen conclusions, particularly the novel demonstration for D2 NAc shell involvement. Likewise, the null result of optically inhibiting D2 inputs to ventral pallidum leaves open the possibility that a more complete or sustained disruption of this pathway may have impaired osPIT.

Conclusions:

The research described here was successful in providing critical new insights into the contributions of NAc D1 and D2 neurons in cue-guided action selection. The authors' data interpretation and conclusions are well reasoned and appropriate. They also provide a thoughtful discussion of study limitations and implications for future research. This research is therefore likely to have a significant impact on the field.

Comments on the previous version:

I have reviewed the rebuttal and revised manuscript and have no remaining concerns.

---

## [Author Response]

The following is the authors’ response to the previous reviews

**Reviewer#1 (Public Review):**
In the current article, Octavia Soegyono and colleagues study "The influence of nucleus accumbens shell D1 and D2 neurons on outcome-specific Pavlovian instrumental transfer", building on extensive findings from the same lab. While there is a consensus about the specific involvement of the Shell part of the Nucleus Accumbens (NAc) in specific stimulus-based actions in choice settings (and not in General Pavlovian instrumental transfer - gPIT, as opposed to the Core part of the NAc), mechanisms at the cellular and circuitry levels remain to be explored. In the present work, using sophisticated methods (rat Cre-transgenic lines from both sexes, optogenetics and the well-established behavioral paradigm outcome-specific PIT - sPIT), Octavia Soegyono and colleagues decipher the diOerential contribution of dopamine receptors D1 and D2 expressing-spiny projection neurons (SPNs).After validating the viral strategy and the specificity of the targeting (immunochemistry and electrophysiology), the authors demonstrate that while both NAc Shell D1- and D2SPNs participate in mediating sPIT, NAc Shell D1-SPNs projections to the Ventral Pallidum (VP, previously demonstrated as crucial for sPIT), but not D2-SPNs, mediates sPIT. They also show that these eOects were specific to stimulus-based actions, as valuebased choices were left intact in all manipulations.This is a well-designed study and the results are well supported by the experimental evidence. The paper is extremely pleasant to read and add to the current literature.

We thank the Reviewer for their positive assessment.

Comments on revisions:We thank the authors for their detailed responses and for addressing our comments and concerns.To further improve consistency and transparency, we kindly request that the authors provide, for Supplemental Figures S1-S4, panels E (raw data for lever presses during the PIT test), the individual data points together with the corresponding statistical analyses in the figure legends.

Panel E of Figures S1-S4 now includes the individual data points. The outcome-specific data have already been analysed, and we report these analyses in the main manuscript. These analyses are more informative than those requested by the Reviewer since they report the net eFects of the stimuli on choice between actions while controlling for potential individual baseline instrumental performance. All data remain fully transparent and are publicly available on an online repository in accordance with eLife policies (see relevant section in Materials and Methods).

In addition, regarding Supplemental Figure S3, panel E, we note the absence of a PIT eOect in the eYFP group under the ON condition, which appears to diOer from the net response reported in the main Figure 5, panel B. Could the authors clarify this apparent discrepancy?

We apologize for the error, which has now been corrected.

We also note a discrepancy between the authors' statement in their response ("40 rats excluded based on post-mortem analyses") and the number of excluded animals reported in the Materials and Methods section, which adds up to 47. We kindly ask the authors to clarify this point for consistency.

We thank the Reviewer for identifying the error reported in our initial response. The total number of animals excluded was 47, as reported in the manuscript.

Finally, as a minor point, we suggest indicating the total number of animals used in the study in the Materials and Methods section.

The total number of animals has been included in the Materials and Methods section.

**Reviewer #2 (Public Review):**
Summary:This manuscript by Soegyono et a. describes a series of experiments designed to probe the involvement of dopamine D1 and D2 neurons within the nucleus accumbens shell in outcome-specific Pavlovian-instrumental transfer (osPIT), a well-controlled assay of cueguided action selection based on congruent outcome associations. They used an optogenetic approach to phasically silence NAc shell D1 (D1-Cre mice) or D2 (A2a-Cre mice) neurons during a subset of osPIT trials. Both manipulations disrupted cue-guided action selection but had no eOects on negative control measures/tasks (concomitant approach behavior, separate valued guided choice task), nor were any osPIT impairments found in reporter only control groups. Separate experiments revealed that selective inhibition of NAc shell D1 but not D2 inputs to ventral pallidum were required for osPIT expression, thereby advancing understanding of the basal ganglia circuitry underpinning this important aspect of decision making.Strengths:The combinatorial viral and optogenetic approaches used here were convincingly validated through anatomical tract-tracing and ex vivo electrophysiology. The behavioral assays are sophisticated and well-controlled to parse cue and value guided action selection. The inclusion of reporter only control groups is rigorous and rules out nonspecific eOects of the light manipulation. The findings are novel and address a critical question in the literature. Prior work using less decisive methods had implicated NAc shell D1 neurons in osPIT but suggested that D2 neurons may not be involved. The optogenetic manipulations used in the current study provides a more direct test of their involvement and convincingly demonstrate that both populations play an important role. Prior work had also implicated NAc shell connections to ventral pallidum in osPIT, but the current study reveals the selective involvement of D1 but not D2 neurons in this circuit. The authors do a good job of discussing their findings, including their nuanced interpretation that NAc shell D2 neurons may contribute to osPIT through their local regulation of NAc shell microcircuitry.

We thank the Reviewer for their positive assessment.

Weaknesses:The current study exclusively used an optogenetic approach to probe the function of D1 and D2 NAc shell neurons. Providing a complementary assessment with chemogenetics or other appropriate methods would strengthen conclusions, particularly the novel demonstration for D2 NAc shell involvement. Likewise, the null result of optically inhibiting D2 inputs to ventral pallidum leaves open the possibility that a more complete or sustained disruption of this pathway may have impaired osPIT.

We acknowledge the reviewer's valuable suggestion that demonstrating NAc-S D1- and D2-SPNs engagement in outcome-specific PIT through another technique would strengthen our optogenetic findings. Several approaches could provide this validation. Chemogenetic manipulation, as the reviewer suggested, represents one compelling option. Alternatively, immunohistochemical assessment of phosphorylated histone H3 at serine 10 (P-H3) oFers another promising avenue, given its established utility in reporting striatal SPNs plasticity in the dorsal striatum (Matamales et al., 2020). We hope to complete such an assessment in future work since it would address the limitations of previous work that relied solely on ERK1/2 phosphorylation measures in NAc-S SPNs (Laurent et al., 2014). The manuscript was modified to report these future avenues of research (page 12).

Regarding the null result from optical silencing of D2 terminals in the ventral pallidum, we agree with the reviewer's assessment. While we acknowledge this limitation in the current manuscript (page 13), we aim to address this gap in future studies to provide a more complete mechanistic understanding of the circuit.

Conclusions:The research described here was successful in providing critical new insights into the contributions of NAc D1 and D2 neurons in cue-guided action selection. The authors' data interpretation and conclusions are well reasoned and appropriate. They also provide a thoughtful discussion of study limitations and implications for future research. This research is therefore likely to have a significant impact on the field.

We thank the Reviewer for their positive assessment.

Comments on revisions:I have reviewed the rebuttal and revised manuscript and have no remaining concerns.

We are pleased to have addressed the Reviewer’s query.

References

Laurent, V., Bertran-Gonzalez, J., Chieng, B. C., & Balleine, B. W. (2014). δ-Opioid and Dopaminergic Processes in Accumbens Shell Modulate the Cholinergic Control of Predictive Learning and Choice. J Neurosci, 34(4), 1358-1369. https://doi.org/10.1523/JNEUROSCI.4592-13.2014

Matamales, M., McGovern, A. E., Mi, J. D., Mazzone, S. B., Balleine, B. W., & BertranGonzalez, J. (2020). Local D2- to D1-neuron transmodulation updates goal-directed learning in the striatum. Science, 367(6477), 549-555. https://doi.org/10.1126/science.aaz5751